# Faunal communities mediate the effects of plant richness, drought, and invasion on ecosystem multifunctional stability

Zhongwang Jing [1,2,4], Jiang Wang[3,4], Yi Bai[3] & Yuan Ge [1,2 ✉]

Understanding the stability of ecosystem multifunctionality is imperative for maintaining ecosystem health and sustainability under augmented global change. However it remains unknown whether and how biological communities mediate multifunctional stability in response to biodiversity loss and disturbances. Here, we conducted a 3-year experiment by exposing 270 plant communities of four plant richness levels, i.e., 1, 2, 4, or 8 species, to drought and exotic plant invasion disturbances. Then, the direct effects of plant richness, drought and invasion, and their indirect effects mediated by the stability of plant, litter-faunal, and soil-faunal communities on multifunctional stability were disentangled. We found that plant richness increased, while drought and invasion decreased ecosystem multifunctional stability, which were mediated by plant or faunal community stability. By incorporating the stability of communities into the complex ecological mechanisms, the completeness and goodness of ecological models for explaining and maintaining the stability of ecosystem multifunctionality will be improved.

[1] State Key Laboratory of Urban and Regional Ecology, Research Center for Eco-Environmental Sciences, Chinese Academy of Sciences, 100085 Beijing, China. [2] University of Chinese Academy of Sciences, 100049 Beijing, China. [3] School of Life Sciences, Taizhou University, 318000 Taizhou, China. [4] These authors contributed equally: Zhongwang Jing, Jiang Wang. ✉email: yuange@rcees.ac.cn

Terrestrial ecosystems comprise multiple components that interact rapidly and systematically to drive the dynamics of multiple ecosystem functions (ecosystem multifunctionality), such as, biomass production, nutrient cycling, carbon stock, and litter decomposition[1,2]. Ecological stability is the key concept describing the responses of these components to disturbances, e.g., stochastic environmental fluctuations, drought, and exotic biotic invasion, especially in the face of augmented global change, laying the basis for understanding and predicting ecosystem dynamics, and maintaining ecosystem sustainability under a changing environment[3,4]. Previous research of ecosystem stability has simply focused on the stability (such as invariability in time and space, and resistance and resilience against disturbances and fluctuations) of individual functions, e.g., productivity stability, but only a few has tried to understand the overall stability of multiple ecosystem functions as a whole, i.e., ecosystem multifunctional stability[5,6] (the specific stability aspect assessed in this study, refers to the invariability (or its similar term reliability[7]) against environmental stochasticity, and the resistance to drought/invasion disturbance), despite the fact that most ecosystems perform multiple functions simultaneously[2,4,8]. Understanding and predicting the ecosystem multifunctional stability in a noisy world, calls for clear awareness of its multidimensional nature (accounting for its multiple function components, and the different aspects of stability against multiple types of disturbances), and requires keen insights into not only the direct biotic/abiotic (e.g., biodiversity and disturbances) controls, but also the indirect cascading effects on multifunctional stability via biological communities across the food web[3,8,9]. Fortunately, pioneering work has already been done to test the effects of biotic/abiotic factors, through the mediating roles of multiple biotas, on multiple types of ecosystem functions[10–14]. However, such knowledge on the regulation of ecosystem multifunctional stability is still lacking[3], especially when the ecosystem is under the control of multiple factors (e.g., plant diversity, drought, and exotic plant invasion) simultaneously. This knowledge gap constrains our understanding of the direct and indirect effects of biodiversity and disturbances on ecosystem multifunctional stability, and thus limits our ability to predict or maintain ecological sustainability under global change[10].

Biodiversity's (e.g., aboveground plant richness) control on ecosystem dynamics has long been a core focus of ecological studies aimed at maintaining or increasing the ecosystem (multi) functionality in the face of global change[2,15–17]. Previous studies suggest that high plant diversity stabilizes the composition of communities of multiple biotas[18–20], as well as individual ecosystem functions (mainly biomass production)[17,21,22]. However, it is still only theoretical whether, and how, plant richness stabilizes ecosystem multifunctionality[5] directly or indirectly by affecting the compositional stability of communities[3,23]. Global change disturbances' (e.g., drought and exotic invasion) regulation on the composition of communities and the fate of ecosystem multifunctionality has also become a central topic of ecological research[15,24–27]. Despite a large body of studies discussing this topic, few have directly looked into the effects of disturbances on the stability of communities or the stability of ecosystem multifunctionality[28,29]. It is also still unclear whether and how disturbances, especially the multiple and entangled ones, affect community compositional stability (e.g., effects of drought/invasion on the invariability, or the resistance to invasion/drought of communities), and what cascading consequences this will have for ecosystem multifunctional stability[30]. Thus, knowledge of, and the mechanisms (e.g., the mediating roles of plant and faunal communities) underlying, the relationship between biodiversity/disturbance and ecosystem multifunctional stability are imperative, particularly in circumstances where multiple factors control and entangle ecosystem dynamics simultaneously.

Plant and faunal communities, both, play key roles in ecosystems[31], linking the dynamics of various ecosystem functions[32,33]. This is due to the plant community's strong ability to shape habitat conditions for communities at higher trophic levels[31], and that faunal communities can act as links between above- and belowground biotas by simultaneously interacting with plants and exerting influences on fungal or bacterial communities[33–35]. Numerous empirical studies have already shown that the dynamics of multiple ecosystem functions, and the effects of biotic/abiotic factors (e.g., plant richness, drought, and exotic plant invasion) on them, largely rely on the communities of multiple biotas, especially those of plants and fauna[10,26,36–38]. Despite, the mediating role of plant or faunal communities, it has rarely been considered in the diversity- or disturbance-ecosystem multifunctional stability relationship models, such that our ability is limited, to understand, predict, or cope with the fate of ecosystem multifunctional stability in the face of climate change. Filling these knowledge gaps requires explicitness about the general patterns of, and mechanisms underlying, richness/disturbance-multifunctional stability relationships.

To test whether, and how, plant richness, drought, and exotic plant invasion will affect ecosystem multifunctional stability by modulating community compositional stability, a 3-year experiment was performed by exposing 270 plant communities, with four levels of plant richness (i.e., 16 monocultures, 10 two-species mixtures, 10 four-species mixtures, 9 eight-species mixtures with 6 replicates in each), to drought (non-, moderate-, and intensive-drought; treated numerically as 0, 1, and 2 during data analyses) and exotic plant invasion (non-invasion and invasion; 0 and 1). We calculated the compositional stability of plants, litter-faunal, and soil-faunal communities based on both their species abundances and phylogenetic traits, and also evaluated the stability of ecosystem multifunctionality based on 14 function-related variables that relate to biomass production (aboveground and belowground plant biomass, light interception efficiency, and litter- and soil-fauna abundances), soil properties (soil carbon, nitrogen, phosphorus, and glomalin related soil protein (GRSP)), or processes (litter decomposition rate, and activities of β-glucosidase, protease, nitrate reductase, and dehydrogenase). Then, we assessed three aspects of stability, i.e., invariability (reliability[7]) against environmental stochasticity, drought resistance, and invasion resistance, for both community stability and multifunctional stability. Finally, the direct and indirect effects of plant richness, drought, and invasion on ecosystem multifunctional stability (all aspects), were assessed using structural equation modelings (SEMs). The overall setup and analyses allowed us to test the following hypotheses: (1) Plant richness increases ecosystem multifunctional stability (Fig. 1a); (2) Drought/invasion decreases multifunctional stability (invariability, and resistance to invasion/drought) (Fig. 1b); (3) Plant, litter-, and soil-faunal community stability mediate the effects of plant richness, drought, and invasion on multifunctional stability (Fig. 1c).

## Results

**Direct effects of plant richness, drought, and invasion on multifunctional stability.** Plant richness, drought, and invasion had significant influences on ecosystem multifunctionality, as evidenced by permutational multivariate analysis of variance (PERMANOVA; $p < 0.001$; Table 1). We further assessed the direct effects (paths derived from SEMs) of the three factors on multifunctional stability and on different stability aspects, i.e., invariability (against stochasticity), drought resistance, and invasion resistance. The results showed that plant richness directly increased, while drought and invasion directly decreased, multifunctional stability ($p < 0.05$, except for drought effect on multifunctional invariability; Fig. 2a invariability, Fig. 2b drought

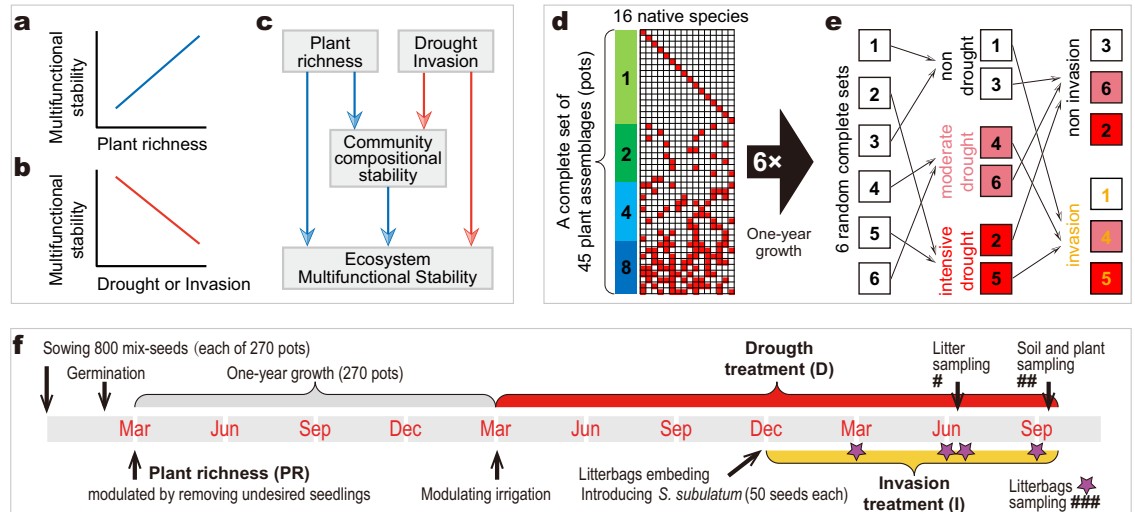

**Fig. 1 Hypotheses and experimental design. a** Plant richness increases ecosystem multifunctional stability (Hypothesis 1). **b** Drought/invasion decrease multifunctional stability (invariability and resistance to invasion/drought) (Hypothesis 2). **c** Plant, litter-, and soil-faunal communities mediate the effects of plant species, drought and invasion on ecosystem multifunctional stability (Hypothesis 3). **d** With the species pool consisting of 16 plant species, the monoculture of each species, and the random mixtures of 2, 4, and 8 species (with 10, 10, and 9 distinct assemblages, respectively) were designed, creating a complete set of 45 distinct plant assemblages. **e** Each plant assemblage was replicated 6 times, for a total of 270 assemblages (pots). After 1-year of growth the assemblages were manipulated with three levels of drought intensities (non-, moderate-, and intensive-drought). Nine months after the introduction of drought stress, 2 complete sets (45 pots) of each drought treatment we randomly exposed (invasion) or not exposed (non-invasion) to the invasive species *Symphyotrichum subulatum*. **f** Experiment schedules. # Litter sampling for litter-fauna extraction. ## Plant sampling for collecting aboveground/belowground plant biomass, and plant community data; Soil sampling for extracting soil-fauna, determining contents of soil organic carbon, nitrogen, phosphorus, and glomalin related soil protein (GRSP), or measuring activities of soil enzymes including β-glucosidase, protease, nitrate reductase, and dehydrogenase; Light interception efficiency was determined monthly from May to August. ### Litterbags sampling for litter decomposition rate evaluation.

**Table 1 Permutational multivariate analysis of variance (PERMANOVA) showing the effects of plant richness, drought, and invasion on communities and multifunctionality.**

| Distances | Factors | $R^2$ (%) | F | p |
|---|---|---|---|---|
| Plant community composition (Weighted UniFrac) | Plant richness | 6.8 | 19.7 | <0.001 |
| | Drought | 0.4 | 1.3 | =0.192 |
| | Invasion | 1.0 | 3.0 | <0.01 |
| Litter-faunal Community composition (Weighted UniFrac) | Plant richness | 0.7 | 2.1 | <0.05 |
| | Drought | 6.9 | 20.5 | <0.001 |
| | Invasion | 0.9 | 2.9 | <0.01 |
| Soil-faunal Community composition (Weighted UniFrac) | Plant richness | 0.7 | 2.0 | =0.061 |
| | Drought | 1.7 | 4.9 | <0.001 |
| | Invasion | 0.9 | 2.0 | =0.067 |
| Multifunctionality (Bray–Curtis) | Plant richness | 4.1 | 13.5 | <0.001 |
| | Drought | 8.9 | 29.1 | <0.001 |
| | Invasion | 4.1 | 13.1 | <0.001 |

influence on the composition of plant and faunal communities (PERMANOVA; $p < 0.05$, except the marginal significant influence that drought exerted on plant, or that plant richness and invasion exerted on soil-fauna ($p < 0.1$; Table 1). As for the direct effects on community stability, the paths derived from the SEMs showed significantly positive effects of plant richness on the stability (all aspects) of plant and soil-faunal communities, and significantly negative effects of drought/invasion on the stability of both litter- and soil-faunal communities ($p < 0.05$; Fig. 2a–c), even after accounting for all side effects.

**Multifunctional stability was positively linked to community stability.** The composition of plant, litter- and soil-faunal communities were significantly related with ecosystem multifunctionality (Mantel test $p < 0.001$; Table 2). For all stability aspects, ecosystem multifunctional stability was significantly and positively linked with the stability of the plant, litter-, and soil-faunal communities ($p < 0.05$; Fig. 2a–c). Moreover, the generally positive links between the stability of individual functions and the stability of communities (Supplementary Table 1) further supported such links observed between community stability and multifunctional stability.

**Plant, litter-, and soil-faunal communities mediate plant richness effect on multifunctional stability.** The results of SEMs based on the three aspects of stability showed that, most of the links along the cascading paths from plant richness—via plant and faunal communities—to multifunctional stability were significantly positive ($p < 0.05$; Fig. 2a–c), the exception being the links from plant richness or plant community to litter-faunal community stability. As a result, significantly positive indirect effects of plant richness on different aspects of multifunctional stability were observed, accounting for 42–51% of the total effects

resistance, and Fig. 2c invasion resistance). Also, the generally positive direct effects of plant richness, and negative direct effects of drought and invasion, on the stability (all aspects) of individual functions (Supplementary Table 1) further supported the observation on multifunctional stability.

**Positive effects of plant richness, and negative effects of drought and invasion on plant and faunal community stability.** Plant richness, drought, and invasion also exerted significant

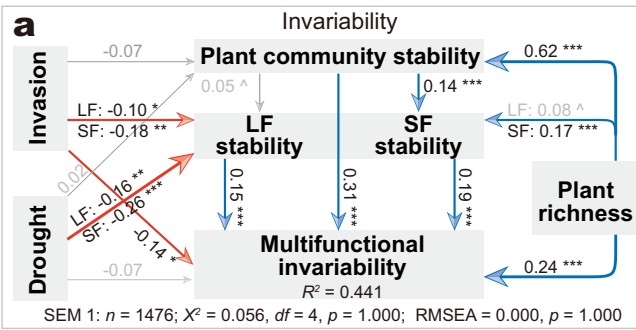

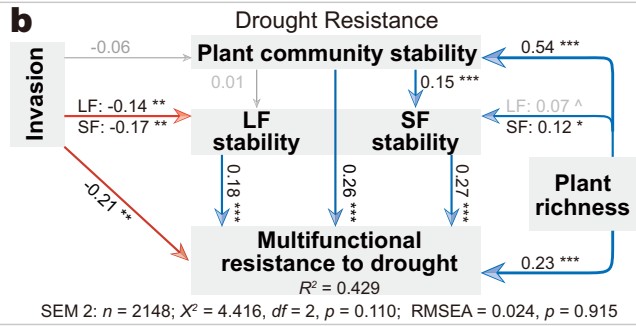

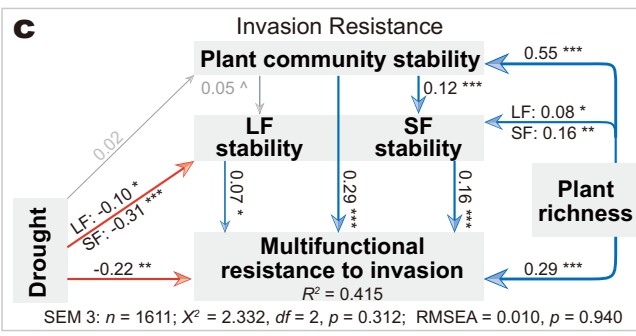

**Fig. 2 Structural equation modelings (SEMs) showing the direct and indirect effects of plant richness, drought, and invasion on ecosystem multifunctional stability. a** Direct and indirect effects of plant richness, drought, and exotic plant invasion on community compositional and ecosystem multifunctional invariability (against stochastic fluctuations), calculated as the pairwise similarity within treatment groups, at same plant richness*drought*invasion condition. **b** Direct and indirect effects on drought resistance calculated as the pairwise similarity between drought (moderate- and intensive-drought) and non-drought treatments, at same plant richness*invasion condition. **c** Direct and indirect effects on invasion resistance calculated as the pairwise similarity between invasion and non-invasion, at same plant richness*drought condition. LF, litter-faunal community; SF, soil-faunal community. n, number of observations (pairwise similarities) extracted from symmetric similarity matrices (Supplementary Fig. 1); $\chi^2$, Chi-square; df, degree of freedom; p, probability level; RMSEA, root-mean squared error of approximation are the goodness-of-fit statistics for each model. Significance of effects derived from SEMs are estimated by conducting a Mantel-like permutation test ($n = 10,000$), and indicated by symbols (*$p < 0.05$, **$p < 0.01$, ***$p < 0.001$, ^$p < 0.1$, and no symbol $p > 0.1$). Blue, positive effect; Red, negative effect.

**Table 2 Mantel test showed the positive relationships among communities, and the significant links between community and ecosystem multifunctionality.**

|  | Spearman-R | p |
|---|---|---|
| Plant vs. Multifunctionality | 0.475 | <0.001 |
| Soil-fauna vs. Multifunctionality | 0.302 | <0.001 |
| Litter-fauna vs. Multifunctionality | 0.202 | <0.001 |
| Plant vs. Soil-fauna | 0.193 | <0.001 |
| Plant vs. Litter-fauna | 0.052 | =0.056 |
| Soil-fauna vs. Litter-fauna | 0.051 | <0.05 |

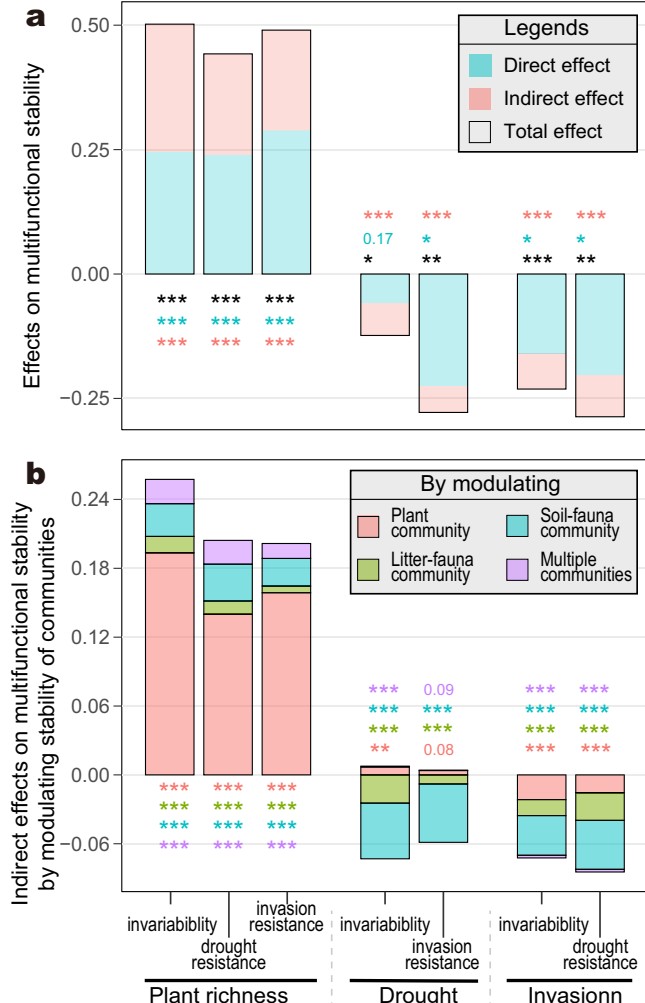

**Fig. 3 Effects of plant richness, drought and invasion on multifunctional stability. a** Total, direct, and indirect effects on multifunctional stability. **b** Indirect effects on multifunctional stability by modulating the stability of plant, litter-, and soil-faunal communities. Stack bars are the unbiased estimates of effects generated by bootstrapping ($n = 10,000$). Asterisk or number near stack bar indicates significance of difference from 0 (decimals $p > 0.05$, *$p < 0.05$, **$p < 0.01$, ***$p < 0.001$; Mantel-like permutation test with 10,000 randomizations).

($p < 0.001$; Fig. 3a). More specifically, when looking into the indirect relative contributions of the three communities to total plant richness effects, we found that plant community accounted for the largest proportion, 33–38%, while litter- and soil-faunal communities accounted for only a small proportion, 1.3–2.8% and 5.1–7.3%, respectively (Fig. 3a, b). In addition, deleting any single path along the mentioned cascading paths (except the path

from plant richness to litter-faunal community in SEM 1; Fig. 2a–c) generated unstable models ($\chi^2$ increased from near 0 to a range between 7.3 and 744.3, $p < 0.05$; Supplementary Table 2), which further supported the importance of the plant and faunal community stability for mediating the positive plant richness effects on multifunctional stability.

**Drought and invasion decreased multifunctional stability indirectly by destabilizing faunal communities.** Drought and invasion had significantly negative indirect effects on different aspects of multifunctional stability by influencing community stability ($p < 0.001$; Fig. 3a). The significantly negative indirect effects of drought and invasion on multifunctional stability were mainly realized by modulating the stability of faunal communities rather than plant community, which was shown by comparing the insignificant paths from drought and invasion to the plant community ($p > 0.05$) with the significant paths linking to faunal communities ($p < 0.05$; Fig. 2a–c), or the lower indirect effects via plant community compared with faunal communities (Fig. 3b). More specifically, indirect effect, by modulating the stability of faunal communities, accounted for 56% (or 21%) of the total drought effects on multifunctional invariability (or invasion resistance), and 21% (or 22%) of the total invasion effect on multifunctional invariability (or drought resistance) (Fig. 3a). Additionally, deleting any single path along the cascading route, from drought or invasion to multifunctional stability, would also destabilize the models ($x^2$ increased to the range between 12.0 and 216.5, $p < 0.05$; Supplementary Table 2), which further supported the importance of stability of faunal communities for the negative effects of drought or invasion on multifunctional stability.

## Discussion

**Evaluation of ecosystem multifunctional stability.** Traditional studies on ecosystem multifunctional stability normally analyze the multidimensional data in a univariate manner, by either reducing data dimensionality (e.g., mean of $z$-scores or first principal component) or calculating similarity in a strict one-to-one correspondence manner. This may tease away undesired noise, but will also simultaneously discard a substantial fraction of information (e.g., the overlap of functional traits among totally different plant species compositions can also drive similar ecosystem functioning, especially for the case of mix-plant stands), leading to a reduction in statistical strength, thus requiring larger sample sizes to generate convincing conclusions on ecological stability. In this study, we analyzed the community and multifunctionality data in a multivariate manner, evaluating the cascading effects of the manipulated factors (plant richness, drought, and invasion) on different aspects (i.e., invariability, drought resistance, and invasion resistance) of community stability or multifunctional stability, using SEMs (Fig. 2) based on the specific aspect of stability extracted from the corresponding subset of a similarity matrix (Supplementary Fig. 1). With these analyses, we were able to assess the links between community stability and multifunctional stability, and the direct/indirect effects of multiple factors on community stability and multifunctional stability, on aspects of invariability, drought resistance, and invasion resistance.

To evaluate the strengths of the multivariate-based SEMs (Fig. 2) over that of the univariate-based ones (Supplementary Fig. 2), we also build the same SEMs by calculating similarity in a strict one-to-one correspondence manner according to Baert et al.[39]. The multivariate-based SEMs (Fig. 2b, c) showed generally comparable levels of statistical significances of effects but higher explained variances of multifunctional resistance to invasion (41.5% versus 30.0%) or drought (42.9% versus 37.2%) compared to the univariate-based ones (Supplementary Fig. 2b, c), suggesting higher strengths of the multivariate-based SEMs over that of the univariate-based ones. Another evidence, that support the higher strengths of the multivariate-based SEMs over the univariate-based ones, is that the former can account for the contribution of the functional overlap between totally different plant species compositions to multifunctional stability especially for the case of high-

diversity communities, as evidenced by the higher effects of plant richness on plant community stability showed by the result derived from the former method (Fig. 2 and Supplementary Fig. 2). Though, both manners also showed similar results, i.e., both a positive plant richness-multifunctional stability relationship and negative disturbance-multifunctional stability relationship can be mediated by the stability of faunal communities (Figs. 2a–c and 3, and Supplementary Fig. 2).

Community stability matters for regulating multifunctional stability. It has been recognized that the composition of communities across the food web play determining roles in regulating the dynamics of various ecosystem functions[38,40,41]. Deeper insights have also been given into the common linkage between ecosystem multifunctional stability and the composition, or more specifically, the compositional stability of communities[3,38,42]. Our results suggested that ecosystem multifunctional stability (invariability, drought resistance, and invasion resistance) was positively linked to the stability of plant, litter-, and soil-faunal communities ($p < 0.05$; Fig. 2a–c), and this result was further supported by the generally positive links between the stability of individual functions and the stability of the three communities (Supplementary Table 1), suggesting that the stability of communities is critical for mediating ecosystem multifunctional stability. Understanding the biotic and abiotic controls on ecosystem stability, requires us to be clear about the cascading link between community stability and multifunctional stability[3,5,9], which are crucial for maintaining and increasing the robustness of ecosystem multifunctionality in the face of global change.

To further address which plant or fauna groups are important to stability of ecosystem multifunctionality, we estimated the relative contributions of plant species (or faunal taxa at the Order level) to the relationship between community stability and ecosystem multifunctional stability (invariability, drought resistance, or invasion resistance) (see supplementary method for detailed estimation procedure). We found that the positive relationship between the stability (all aspects) of community (plant, soil fauna, and litter fauna) and multifunctionality decreased with the remove of plants species (or faunal Orders) from the original community, and the level of such decrease increased with the number of plants species (or faunal Orders) removed (Supplementary Fig. 3), demonstrating that the co-working and interaction among multiple plants (or fauna) are important for the maintenance of multifunctional stability, despite that some dominant plant species (*Patrinia scabiosifolia, Artemisia stolonifera*, and their combinations with others) or faunal Orders (soil fauna: *Posuromorpha* and *Trombidiformes*; litter fauna: *Trombidiformes* and *Araneae*) accounted for relatively high level of contributions (Supplementary Fig. 4). We also found that the higher the mean abundances of the plant species or faunal taxa that drive the ecosystem functions, the greater their contribution to multifunctional stability (Supplementary Fig. 5a–c). In addition, the contribution of plant species (or their combinations) to the community-multifunctional stability decreased with the mean coefficient of variance (CV) of biomass, while those of soil and litter-faunal Orders (or their combinations) increased with the mean CV of faunal abundance (Supplementary Fig. 5d–f), suggesting that the stable biomass production of dominant plant species while the asynchronous dynamics of faunal taxon that drive the dynamic of ecosystem functions were also key to shaping the community-multifunctional stability relationship in the face of disturbances (spatial stochasticity, drought and invasion).

Plant richness and disturbances modulate community stability. Our results showed that higher plant richness supported higher plant and faunal community stability, for the aspects of invariability, drought resistance, and invasion resistance (Figs. 2a–c and 3a). This

result is supported by the diversity-stability hypothesis that community stability increases with species richness[43], and is consistent with the ample empirical evidence present in the literature that shows plant richness stabilizes communities across trophic levels[18,20]. Several mechanisms, notably asynchronous performance associated with mix-species plant communities and compensatory dynamics generated by negative species covariance and selection for stable dominant species populations, may underlie the positive richness-community stability relationship[44]. Moreover, the bottom-up regulation of a plant community on soil organisms has also been suggested to be the indirect mechanism driving the stabilizing effects of plant richness on communities[10], because plant communities can strongly shape the habitat for communities of higher trophic levels[45,46] as evidenced by the positive relationship between the communities of plant and soil-fauna ($p < 0.001$; Table 2).

Both drought and plant invasion had neutral effects on the stability of the aboveground plant community, but significantly decreased that of both litter- and soil-faunal communities, for different aspects of stability (Fig. 2a–c). This suggests that drought and exotic plant invasion may have a more prominent direct effect on litter- and soil-faunal communities than on aboveground plant communities, which may imply a top-down governance on the effects of drought or invasion on communities[46]. The destabilizing effects of disturbances, e.g., drought and exotic invasion, on community stability may be governed by multiple mechanisms relating to the increase of environmental heterogeneity[47], the impairment of local asynchrony and insurance effect[48], the decrease of belowground competition[49], the reduction of species movement and dispersion[26], and the decoupling of multitrophic interactions[30,46]. In addition, faunal communities occupy a key position within the soil food web, as they interact intensively with the plant assemblage and influence the fungal and bacterial communities simultaneously[26,34,45], thereby linking the dynamics of ecosystem functions[32,36,50]. Thus, the indirect role of faunal communities in modulating multifunctional stability should be considered to gain a holistic understanding of the ecosystem multifunctional stability under climate change[10,26].

Plant richness increases multifunctional stability by increasing community stability. There was a significantly positive effect of plant richness on multifunctional stability, for different stability aspects ($p < 0.01$; Figs. 2a–c and 3a). Plant richness effect on the ecosystem stability highly depends on the dynamics of the composition and functional traits of plant community[51], as well as multitrophic interactions[52]. Several mechanisms, including insurance effect, portfolio effect, and compensatory interactions, have been proposed to underpin the effect of plant richness on ecosystem stability[53–56]. These mechanisms highlight the higher probability to include more disturbance-tolerant species[30], diversified ecological traits and niches, and the intensified interspecific interactions within mix-plant stands, which allow compensatory performances to buffer community compositional and functional variation against disturbances or fluctuations[52].

Community composition has been suggested to play an even more important role than biodiversity per se in driving the dynamics of multiple ecosystem functions[40,41,57,58], and this lays the basis for the logic that models of plant richness-multifunctional stability relationship may be improved by incorporating the compositional traits (e.g., stability) of communities across the food web. Our results based on SEMs showed the positive indirect effects of plant richness on multifunctional stability by increasing the stability of plant, litter-, and soil-faunal communities (Figs. 2a–c and 3), with the indirect effects accounting for 42–51% of the total plant richness effects ($p < 0.01$; Fig. 3a). This suggests the significant role of community stability in mediating the plant richness-multifunctional stability relationship. Since only a small fraction of the biotas (i.e., plant and faunal communities) across the food web were included in this study, substantial parts of plant richness effects still cannot be explained by stability of the communities (reflected as the direct plant richness effects; Figs. 2a–c and 3a). Thus, we propose that, by exploring the stability of adequate communities across multiple trophic levels with the plant richness-multifunctional stability models, the completeness and goodness of ecological models for explaining, maintaining, and predicting plant richness-multifunctional stability relationship would be improved.

Drought and invasion decrease multifunctional stability by decreasing the stability of litter- and soil-faunal communities. Our results also showed a negative disturbance-multifunctional stability relationship (Fig. 3a). Among the large body of studies on ecosystems under global change disturbances, only a few have looked into the effects of disturbances on ecosystem functional stability, e.g., drought and invasion reducing productivity stability and ecosystem resistance[28,29]. Thus, knowledge on the general ecological pattern of disturbance-multifunctional stability relationship and the underlying mechanisms is imperative, particularly in cases where compounded disturbances occur simultaneously. However, the cascading links from disturbances —through multitrophic communities—to numerous ecosystem functions, for example, the faunal community compositional change induced by plant invasion was associated with soil nitrogen cycling[59], may help us to take a snapshot of potential general pattern of disturbance-multifunctional stability relationship.

Drought and invasion significantly decreased multifunctional stability by decreasing litter- and soil-faunal community stability ($p < 0.05$; Figs. 2a–c and 3a, b), and the indirect effects accounted for more than 20% of the total effects ($p < 0.01$; Fig. 3a), suggesting significant roles for faunal communities in mediating the disturbance-multifunctional stability relationship. Similarly, the direct disturbance effects (Figs. 2a–c and 3a) may be further explained by the stability of biological communities that were not assessed in this study. More insights into the stability of multitrophic communities mediating the disturbance-multifunctional stability models are imperative, for understanding and predicting ecosystem multifunctional stability in the face of intensified global change under circumstances where multiple disturbances stress ecosystems.

Our results showed a significantly positive plant richness-multifunctional stability relationship and a significantly negative disturbance-multifunctional stability relationship. To the best of our knowledge, this is the first study that treated and incorporated plant richness, drought, and invasion as influencing factors into a single ecological model to assess the indirect mechanism(s), by modulating community stability, underlying the ecological patterns of the plant richness- and disturbance-ecosystem multifunctional stability relationships, and newly evidenced that both relationships can be mediated by the stability of plant or faunal communities. Our results might be strengthened by including the temporal aspects of stability (e.g., temporal stability and recovery of community and multifunctionality)[5,6]; however, constrained by the non-time-series dataset, the temporal aspects of stability can't be assessed in this study. Further studies should incorporate more aspects of stability to build a more general framework for understanding and predicting the diversity- and disturbance-multifunctional stability relationships. Nevertheless, on the aspects of invariability, resistance of both community and ecosystem multifunctionality, the loss of plant diversity, and the drought and invasion distrubances would likely reduce the sustainability of ecosystem multifunctionality in the face of global change.

## Methods

**Design.** *Plant richness.* Sixteen locally frequent native plant species in the barren mountain areas (around Taizhou University, Zhejiang, China) invaded by the exotic plant *Symphyotrichum subulatum*[60] were selected as the native species pool. These species were chosen because they spanned the dicotyledon plant taxonomy (including 7 Orders, 10 Families, and 14 Genus, in the Class *Magnoliopsida*), differed widely in their functional traits (related to height, life form, dominance in local communities, and leaf habit) (Supplementary Table 3), and were occasionally found to be associated with the invasive species *Symphyotrichum subulatum*[60] in the local secondary-succession communities. With this species pool, we were able to imitate the locally natural, spatially stochastic, compositionally ruderal, and functionally varied plant community[61], which is a typical attribute of the secondary-succession communities in the local barren mountains invaded by the exotic plant *Symphyotrichum subulatum*. Based on this native species pool, monocultures of each species (16 total), and random mixtures of 2, 4 or 8 species (with 10, 10, or 9 distinct assemblages, respectively) were designed, creating a complete set (Fig. 1d) of 45 different plant assemblages (pots) in total. Each plant assemblage was replicated 6 times, for a total of 270 pots. To eliminate the non-random effects during the 1-year development of the 270 pots, their distributions were randomized, such that not all replicates of an assemblage were next to each other (Fig. 1d–f).

*Drought.* After 1-year development of the native plant assemblages, three drought treatments (non-, moderate-, and intensive-drought) were manipulated by adjusting irrigation using automatic drip irrigation systems, with 100%, 50%, and 25% of the equivalent to the amount received in the areas where native species were collected, respectively. Two random complete sets were selected for each drought treatment, each complete set being composed of 45 different plant assemblages (Fig. 1d–f).

*Exotic plant invasion.* Nine months after drought treatment, the two complete sets (Fig. 1d) of each drought treatment were randomly exposed (invasion) or not exposed to (non-invasion) the invasive species *Symphyotrichum subulatum* (Michx.) G. L. Nesom (Fig. 1e, f). *S. subulatum*, an annual herbaceous plant native to North America, is a common invasive species in the subtropical and tropical regions of China[18,60], and tends to interact with the native species via, for example, competing for space and resources[62,63], enriching for pathogens or herbivores, and changing soil faunal, bacterial or fungal microbiomes[18,64,65].

**Experiment.** The experiment based on the design mentioned above was conducted at Taizhou University, Zhejiang province, China (28.66°N, 121.39°E). The seeds of the 16 native plant species (Supplementary Table 3) and the soil were collected from nearby mountain areas (Wugui, 28.65°N, 121.38°E; Baiyun, 28.67°N, 121.42°E; Beigu, 28.86°N, 121.11°E). The seed-mixtures were obtained by mixing seeds of the 16 species pro rata, in proportion to germination rates. The soil (fine-loamy, mixed, semiative, mosic, Humic Hapludults) was sieved to pass a 2-mm mesh, and thoroughly mixed. 270 plastic pots (72 cm length × 64 cm width × 42 cm depth) were prepared, and each was filled with a 27-cm soil layer, followed by a 10-cm mixture of soil and vermiculite-compost to provide water-, air- and fertility-support for germination, seedling establishment, and plant growth (Supplementary Table 4).

*Native plant assemblages.* All the 270 pots were placed inside a plastic shelter, which allowed for both air ventilation and protection from rain. Each pot was sown with a seed-mixture of ca. 800 seeds. One month after germination, for each pot, the undesired seedlings were removed manually according to the plant richness design (Fig. 1d–f), and thus 32 vigorous seedlings (with the same number of seedlings per species, e.g., 4 seedlings for each species of the 8-species mixtures) were spatial-evenly retained. In this manner, the plant richness was manipulated for each plant assemblage. During the development of the 270 plant assemblages, the soil volumetric water content was controlled at ca. 20%, which was similar to that of the nearby mountainous soil, using the automatic drip irrigation systems. Weeds and undesired species were removed monthly (Fig. 1f).

*Drought treatment.* After 1-year development of native plant assemblages, the drought treatments (non-, moderate-, and intensive-drought) were manipulated according to the experimental design mentioned above (Fig. 1d, e). Two complete sets (Fig. 1d) of different plant assemblages (2 × 45 pots) were selected for each drought treatment. Every other week, 40 pots each drought treatment were randomly selected for measuring soil water content and soil temperature at the depth of 0–20 cm, using the ProCheck analyzer (Decagon, Pullman, Washington, USA), and irrigation was adjusted accordingly using automatic drip irrigation systems. The irrigation for non-, moderate-, or intensive-drought was adjusted to accomplish an irrigation level amounts to 100%, 50%, or 25% that of the mountain areas where seeds were collected. Because of the distinct seasonal temperature and evaporation conditions, the irrigation frequencies were approximately daily in May-September, every other day in March–April and October–December, and weekly in January–February. With this manipulation, the volumetric soil water contents of non-, moderate-, and intensive-drought were controlled within ranges

of 13.8–23.4%, 6.8–13.7%, and 1.4–7.4%, respectively, throughout the manipulation of drought treatment (Fig. 1e, f). Eight months after drought introduction, fresh litter was collected form the two replicate pots of each drought treatment, and then oven-dried at 40 °C, cut into ca. 2-cm pieces, and filled into litterbags (2-g litter in each litterbag).

*Invasion treatment.* Nine months after drought introduction, one complete set (45 pots) of the plant assemblages (Fig. 1d) from each drought treatment, was chosen and exposed to invasion disturbance by sowing 50 seeds of *S. subulatum* in each pot, and the other was specified as the non-invasion treatment (Fig. 1e, f). The prepared litterbags were embedded under the litter-layer of each pot (5 litterbags in each pot), correspondingly.

**Sampling.** Six months after invasion introduction, one litterbag was collected for litter-fauna extraction. Nine months after invasion, five soil cores (20-cm depth) were collected with augers (6.4 cm in diameter) and mixed for extraction of soil-fauna, and measurement of soil property and enzyme activity (Fig. 1f). The aboveground biomass of both native and invasive plants in each pot was harvested, sorted to species, oven-dried to a constant mass at 80 °C, and weighed. The belowground plant biomass was also sampled, sorted to native and invasive groups, oven-dried, and weighed (Fig. 1f).

### Plant, litter-, and soil-faunal communities

*Plant community.* Since exotic plant invasion was treated as a disturbance factor, the biomass of the invasive species *S. subulatum* was not included for analyses concerning plant community and ecosystem (multi)functionality. The aboveground biomasses of native plant species in each of the 270 pots were collected for plant community analysis.

*Litter- and soil-faunal communities.* One litterbag or fifty grams of mixed-soil samples were used for litter- or soil-fauna extraction using a Tullgren funnel apparatus (dry funnel method)[66]. The obtained microarthropods were stored in 70% alcohol, identified with double-tube anatomical lens, and classified to Family level. For both litter and soil samples, the numbers (abundances) of all faunal taxa were counted for litter/soil-faunal community analysis.

**Phylogenetic information of plant, litter-, and soil-faunal communities.** Similar procedures were used to construct the plant and faunal phylogenetic trees. First, protein sequences of 12 faunal mitochondrial coding genes and 16 plant plastid coding genes (Supplementary Data 1) were obtained by searching plant or faunal taxonomies from NCBI protein database (https://www.ncbi.nlm.nih.gov/protein/) with Edirect software (https://www.ncbi.nlm.nih.gov/books/NBK179288/). All available sequences at plant species level or faunal Family level were fetched. If unavailable, the missing sequences were sampled from plant genus or faunal Order level. *Sequoiadendron giganteum* and *Echinococcus* were specified as out-group references for plant and faunal trees, respectively. Then, the sequences of each plant or faunal taxon were clustered at 97% or 90% identity independently, and the centroids were used as representative markers. The markers were aligned with MUSCLE[67], followed by concatenation. Finally, using MEGA X[68], the maximum likelihood trees were constructed based on BioNJ initial trees[69] and 500 bootstrap checking nodal support. The parameters for plant tree construction were specified as follow: 70% partial deletion (with 4824 positions retained) and the best-fit substitution model JTT + G + I + F[70,71]; parameters for faunal tree: 90% partial deletion (2778 positions) and LG + G + I + F model[71,72]. The Linux codes for processing the protein sequences were submitted to GitHub (https://github.com/YuanGe-Lab/JZW_2022/tree/main/linux)The plant and faunal taxonomies, representative markers, and marker accessions are provided as Supplementary Data 1.

**Ecosystem function-related variables.** A total of 14 individual function-related variables were collected. These variables belonged to three functional groups: (1) biomass production, including aboveground and belowground biomass of native plants, light interception efficiency, litter-fauna abundance, and soil-fauna abundance; (2) soil properties, including contents of soil organic carbon, soil nitrogen, soil phosphorus, and GRSP (relating to soil physical properties and stocks of carbon and nutrient[73]); and (3) processes, including rate of litter decomposition, and activities of β-glucosidase, protease, nitrate reductase and dehydrogenase.

Light interception efficiency, the fraction of incident photosynthetically active radiation (PAR) intercepted by each plant community canopy, was determined between 12:00 and 14:00 on clear days using LI-191R line PAR sensors (LI-COR Inc., NE, USA), and the mean of 4 measurements (monthly from May to August the third year; Fig. 1f) was used. Total soil organic carbon and nitrogen were measured with an elemental analyzer (vario Max; Elementar, Germany). Total soil phosphorus was determined using the molybdenum blue method with a UV–visible spectrophotometer (Shimadzu, Kyoto, Japan). GRSP was determined using the method described by Shen et al.[18]. Litter decomposition rate was assessed by embedding litterbags and fitting litter mass loss against decomposition time (Fig. 1f). Enzyme activities were analyzed by the spectrophotometric method using

the substrates, p-Nitrophenyl-β-ᴅ-glucopyranoside (pNPG; for β-glucosidase), caseinate (protease), nitrate (nitrate reductase) and triphenyltetrazolium chloride (TTC; dehydrogenase)[18].

**Quantifying community stability and multifunctional stability.** Community data was comprised of native plant biomasses or faunal abundances, and the associated phylogenetic information. Multifunctionality data was comprised of 14 function-related variables, each variable ($V$) being transformed ($V'$) using the formula $V' = \frac{V - \min(V)}{\mathrm{sd}(V)}$ to guarantee even contribution to global variance. We calculated community similarity (1 minus Weighted-UniFrac distance) and multifunctional similarity (1 minus Bray–Curtis distance), based on the community data and the multifunctionality data, respectively. The specific subsets of each symmetric similarity matrix were used to assess three different aspects of stability: (1) Invariability (against stochastic fluctuations), reflected as the pairwise similarities (1476 pairs) within treatment groups, at same plant richness*drought*-invasion condition; (2) Drought resistance, the similarities (2148 pairs) between drought (moderate- and intensive-drought) and non-drought treatments, at same plant richness*invasion condition; and (3) Invasion resistance, the similarities ($n = 1611$ pairs) between invasion and non-invasion treatments, at same plant richness*drought condition (Supplementary Fig. 1).

We also assessed the three aspects of stability of each individual function in a similar way, but by calculating the similarity using the formula $\mathrm{SIM}_{ij} = 1 - \frac{|V_i - V_j|}{V_i + V_j}$ ($V_i$ and $V_j$ are $i$th and $j$th elements in a function vector; $\mathrm{SIM}_{ij}$ is the similarity between $V_i$ and $V_j$).

**Statistics and reproducibility.** PERMANOVA (10,000 randomizations) was conducted to test the influences of the manipulated factors on ecosystem multifunctionality or communities of plant, litter- and soil-fauna, using "vegan::adonis" in R[74]. Mantel test (10,000 randomizations; *Spearman's R*) was conducted to test the community-community or the community-multifunctionality relationships, using "vegan::mantel" in R[74].

As each similarity-pair of each aspect of community or multifunctional stability mentioned above was in strict correspondence to single level of each manipulated factor (plant richness, drought, and invasion) (Supplementary Fig. 1), the direct/indirect effects of treatments on the community or multifunctional stability can be assessed using SEM. To test direct and indirect effects (by modulating community stability) of the manipulated factors on multifunctional stability, we built three SEMs (Fig. 1a–c) based on three different aspects of stability (i.e., invariability, drought resistance, and invasion resistance) under the conditions of corresponding parings of manipulated factors (Supplementary Fig. 1), with the LAVAAN package[75]. The standardized paths (direct effects) in SEMs can be conceived as the partial correlations after teasing all side effects away. Bootstrapping with 10,000 randomizations was conducted to generate the unbiased mean effect. The significance of effect was tested using a Mantel-like permutation (10,000 randomizations) test[76], where the null hypotheses (H0) were that the independent factors plant richness, drought, and invasion, had no direct/indirect effects (effect = 0) on multifunctional stability. Based on H0, permutation procedure was conducted by permuting the index of dependent factors (both columns and rows of a symmetric matrix; Supplementary Fig. 1) simultaneously to gain null models and null effects. *p*-values (probability of H0 acceptance) were calculated as the percentage of observed positive (or negative) effect that was greater (or less) than the null effects. We also assessed the direct and indirect effects of factors on the stability of each individual function based on the same SEMs, to consolidate our findings on multifunctional stability. The R codes and examples solving the permutation test for the significance of effects derived from SEMs that based on multidimensional similarity (or distance) were submitted to GitHub (https://github.com/YuanGe-Lab/JZW_2022/tree/main/R). All the analyses were conducted using R (https://www.r-project.org).

**Reporting summary.** Further information on research design is available in the Nature Research Reporting Summary linked to this article.

## Data availability

The datasets generated and/or analyzed during the current study are available at GitHub (https://github.com/YuanGe-Lab/JZW_2022/tree/main/R/data_files).

## Code availability

The executable codes that support our findings, figures and tables are available at GitHub (https://github.com/YuanGe-Lab/JZW_2022/tree/main/R).

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

## Acknowledgements

This work was supported by the Second Tibetan Plateau Scientific Expedition and Research Program (2019QZKK0308 and 2019QZKK0306), the National Natural Science Foundation of China (42177274 and 31870504), and the State Key Laboratory of Urban and Regional Ecology (SKLURE2022-1-3).

## Author contributions

J.W. and Y.G. conceived the ideas and designed the experiment; J.W. and Y.B. collected the data; Z.J. and Y.G. analyzed the data, and led the writing of the manuscript. All authors contributed critically to the drafts. Z.J. and J.W. contributed equally to the work.

## Competing interests

The authors declare no competing interests.
