## [Peer Review File · Communications Biology]

Reviewers' comments:

Reviewer #1 (Remarks to the Author):

<Comments for the authors COMMSBIO-21-2959-T>

This study conducted a pot experiment where plant species richness was manipulated by sowing (1, 2, 4, 8 species). Different species richness treatments were then subjected to drought and invasion manipulations. Based on the data, the authors focused on the similarities among plant and soil-faunal communities, and ecosystem multifunctionality to disentangle the effects of plant species richness, drought, and invasion on the stability of ecosystem multifunctionality. The text is well written, and the topic is of highly concern in the current biodiversity-ecosystem multifunctionality understanding. However, I have two major concerns. First, based on my understanding, the data analyzed in this study does not have temporal aspects (i.e., variability across multiple time scales, or changes from pre-disturbance to post disturbance states). With this in mind, I am not convinced that the study properly disentangled the effects of plant species richness on "ecosystem multifunctional stability". Second, the way how similarities in plant and faunal composition and ecosystem multifunctionality were calculated is not very clear to me. For example, if the authors would like to quantify differentiations of multiple functions under disturbed conditions from those under control conditions as stability of ecosystem multifunctionality, similarities/dissimilarities need to be calculated as compositional changes from control conditions (or did I misunderstand the text?...if so, please explain more about this point). If these concerns make sense, please revise throughout the text and a bit tone down the issues regarding ecosystem multifunctional stability.

<Specific comments and suggestions>

1. Lines 52-54. "few has tried to understand the overall stability of multiple ecosystem functions as a whole, i.e. ecosystem multifunctional stability". Stability of ecosystem multifunctionality is understudied, but there are a few previous studies. As such, I suggest authors to cite such studies here and elsewhere relevant; Sasaki et al. (2019) *Journal of Ecology*, Wagg et al. (2021) *eLife*.
2. Line 111. "1, 2, 4 and 8 species, with 10, 10, and 9 replicates)". Based on the descriptions in the Methods, this description is not correct (i.e., 16 monocultures, 10 two-species mixtures, 10 four-species mixtures, 9 eight-species mixtures with 6 replicates in each). Please revise.
3. Lines 237-242. This statement is repetition of the statement described in the second paragraph of the previous section (Lines 220-228). Throughout the Discussion, I suggest authors to reduce repetitive descriptions as far as possible, as I feel the Discussion section is considerably lengthy to its content. Readers will be missing the core messages from this study.
4. Lines 482-493. I am not very sure how the authors incorporated distance-matrices data of plant and faunal community composition and ecosystem multifunctionality into the SEMs. I suggest the authors to describe more appropriate explanations.
5. This study focused on plant and faunal community composition and compositional stability as mediators in the effects of plant species richness on ecosystem multifunctionality. As such, I am curious about which species or species groups are important to stability of ecosystem multifunctionality. This point is not discussed well in the Discussion.

Reviewer #2 (Remarks to the Author):

Community Biology

Manuscript Number: COMMSBIO-21-2959-T

Faunal communities mediate the effects of plant richness, drought, and invasion on ecosystem multifunctional stability

The authors collected extensive data for three years to investigate the effects of plant richness, drought and invasion, and their indirect effects on multifunctional stability. The data is valuable. However, I have major concerns on the experiments and statistical framework which might lead to false conclusion. I hope the authors can address them and improve the current work.

Major concerns:

1. In my opinions, the native species selection is always biased based on availability and easiness of experiment setup. However, the selection of native species can affect the results and interpretation greatly. Hence, the reason of the selection cannot just be they are common species there. This would be the first flaw of the study. The authors need to document the attributes and functional traits of native species and their own interactions. The selection criteria should be included. Moreover, how these species could interact with invasive species? The invasive species might not be "invasive" enough for some native species. The document of the interaction between the invasive and native species is weak.

2. The experiment is impressive. However, following the comment above, random mixtures of 2, 4, or 8 species can introduce different levels of noise into your work. The authors did not implement any correction for removing the noise. Moreover, there are 10, 10, or 9 distinct assemblages. The authors did not state the reason why assemblages were not designed as 10, 10, or 10.

3. The data analyses are not appropriate. The authors introduced the Mantel test but lacked a clear statistical framework specifying the null and alternative hypotheses. For example, the tests can be flawed in the presence of autocorrelation and return erroneously low p-values (Reference: doi: 10.1111/2041-210x.12018)

Given the reasons above, I would not be so confident with the results. Indeed, the results are rather intuitive and match our common understanding of community ecology. I wonder what the novel or most important findings are here.

The section of Discussion is disappointing because the authors only compared some of their findings to others. The scope is rather narrow. Authors also failed to document the caveats and potential improvement of the study.

Responses to Reviewer's Comments

Your manuscript entitled "Faunal communities mediate the effects of plant richness, drought, and invasion on ecosystem multifunctional stability" has now been seen by 2 referees. You will see from their comments below that while they find your work of considerable interest, some important points are raised. We are interested in the possibility of publishing your study in Communications Biology, but would like to consider your response to these concerns in the form of a revised manuscript before we make a final decision on publication.

The responses to the Reviewers' comments are provided herein. We greatly appreciate the opportunity to respond and have changed the manuscript in accordance with our responses. In the following responses to the Reviewers' comments, the original comments are italicized, and the responses are indented and in regular font.

We therefore invite you to revise and resubmit your manuscript, taking into account the points raised. In particular,

1) please explicitly address the concerns regarding experimental design and data analyses, as raised by the second reviewer.

To address the second Reviewer's concerns, we have detailed the selection criteria for the native species pool and explained the appropriateness of the mantel-like permutational test for the analyses of the current study. Please see our replies to the second Reviewer's comments.

2) please explicitly address how the current study can disentangle the effects of plant species richness on "ecosystem multifunctional stability"

Please see our responses to the major comment raised by the Reviewer 1.

3) please clarify how similarities in plant and faunal composition and ecosystem multifunctionality were calculated in the current study

Please see our responses to the major comment and minor comment #4 raised by the Reviewer 1.

4) please justify the selection criteria of the native species selection in the current study

Please see our response to the first comment raised by the Reviewer 2.

Reviewers' comments:

Reviewer #1 (Remarks to the Author):

This study conducted a pot experiment where plant species richness was manipulated by sowing (1, 2, 4, 8 species). Different species richness treatments were then subjected to drought and invasion manipulations. Based on the data, the authors focused on the similarities among plant and soil-faunal communities, and ecosystem multifunctionality to disentangle the effects of plant species richness, drought, and invasion on the stability of ecosystem multifunctionality. The text is well written, and the topic is of highly concern in the current biodiversity-ecosystem multifunctionality understanding.

We thank the Reviewer for pointing out the significance of our study and the overall positive comments.

However, I have two major concerns. First, based on my understanding, the data analyzed in this study does not have temporal aspects (i.e., variability across multiple time scales, or changes from pre-disturbance to post disturbance states). With this in mind, I am not convinced that the study properly disentangled the effects of plant species richness on “ecosystem multifunctional stability”. Second, the way how similarities in plant and faunal composition and ecosystem multifunctionality were calculated is not very clear to me. For example, if the authors would like to quantify differentiations of multiple functions under disturbed conditions from those under control conditions as stability of ecosystem multifunctionality, similarities/dissimilarities need to be calculated as compositional changes from control conditions (or did I misunderstand the text?...if so, please explain more about this point). If these concerns make sense, please revise throughout the text and a bit tone down the issues regarding ecosystem multifunctional stability.

Due to its multidimensional nature, ecosystem multifunctional stability comprises multiple stability aspects, such as the invariability against environment stochasticity, resistance against various disturbances, and temporal variability across multiple time scales and resilience/recover from pre-disturbance to post-disturbance states (the temporal aspects of stability as Reviewer concerned). Constrained by the non-time-series dataset, the temporal aspects of stability can't be assessed in this study. But, instead, the other three specific stability aspects were assessed in this study, including the invariability (or its similar term reliability (Naeem and Li 1997)) against environmental stochasticity, the

resistance to drought, and the resistance to invasion. We agree with the Reviewer that these three aspects of stability couldn't totally represent the full aspects of the stability as indicated by the term “multifunctional stability”, and thus we revise our manuscript by parenthesizing the specific aspect of stability assessed in current study after the term “multifunctional stability” at its first appearance in the Introduction, Result, and Discussion sections (please see **Lines 55-56, 140-142 and 215-216**). Also, we added some sentences in the Conclusion section to address such issue as the Reviewer concerned and the necessity to include more aspects of stability with future study/design to build a more general framework for understanding and predicting the diversity- and disturbance-multifunctional stability relationships. Please see **Lines 371-377**.

Fig. S1 An example (plant richness = 1) showing the subsets of each symmetric similarity matrix used for assessing three different aspects of stability (invariability, drought resistance and invasion resistance). I0, non-invasion; I1, invasion; D0, non-drought; D1, moderate-drought; D2, intensive-drought. The subset data for each of other levels of plant richness (2, 4, or 8 species) were extracted similarly. Data marked by the dashed line were those in strict one-to-one correspondence manner and used for generating **Fig. S2**.

We are sorry for the unclear description of the calculation for the similarities used to assess different aspects of stability. The specific subsets of each symmetric similarity matrix were used to assess three different aspects of stability: 1) Invariability (against stochastic fluctuations), reflected as the pairwise similarities (1476 pairs) within treatment groups, at same plant richness*drought*invasion condition; 2) Drought resistance, the similarities (2148 pairs) between drought (moderate- and intensive-drought) and non-drought treatments, at same plant richness*invasion condition; and 3) Invasion resistance, the similarities (n=1611 pairs) between invasion and non-invasion treatments, at same plant richness*drought condition. To help the Reviewer and the readers to understand

how the specific aspect of stability was calculated we supplied an example for the plant richness level of the monoculture treatment as **Fig. S1**. Please see **Lines 527-537** and **Fig. S1**.

1. *Lines 52-54. “few has tried to understand the overall stability of multiple ecosystem functions as a whole, i.e. ecosystem multifunctional stability”. Stability of ecosystem multifunctionality is understudied, but there are a few previous studies. As such, I suggest authors to cite such studies here and elsewhere relevant; Sasaki et al. (2019) Journal of Ecology, Wagg et al. (2021) eLife.*

Thanks for the Reviewer’s notification of these two important citations. We have checked throughout the manuscript and cited these studies here and elsewhere relevant. Please see **Lines 54, 79, and 373**.

2. *Line 111. “1, 2, 4 and 8 species, with 10, 10, and 9 replicates)”. Based on the descriptions in the Methods, this description is not correct (i.e., 16 monocultures, 10 two-species mixtures, 10 four-species mixtures, 9 eight-species mixtures with 6 replicates in each). Please revise.*

We are grateful for the Reviewer’s careful reading and helpful correction on this sentence. This sentence has been revised as suggested. Please see **Lines 112-113**.

3. *Lines 237-242. This statement is repetition of the statement described in the second paragraph of the previous section (Lines 220-228). Throughout the Discussion, I suggest authors to reduce repetitive descriptions as far as possible, as I feel the Discussion section is considerably lengthy to its content. Readers will be missing the core messages from this study.*

Thanks for the suggestion. We have made a careful check throughout the discussion section, and deleted some redundant sentences for clarity. Please see **Lines 222-238, 244, 293, 321-322, 330 and 351-352**.

4. *Lines 482-493. I am not very sure how the authors incorporated distance-matrices data of plant and faunal community composition and ecosystem multifunctionality into the SEMs. I suggest the authors to describe more appropriate explanations.*

Because each similarity-pair of each aspect of the community or multifunctional stability (Please refer to our response to the Reviewer’s major comment) mentioned above was in strict correspondence to single level of manipulated

factor (plant richness, drought, and invasion) (**Fig. S1**), the direct/indirect effects of treatments on the community or multifunctional stability can be assessed using SEM. To test direct and indirect effects (by modulating community stability) of the manipulated factors on multifunctional stability, we built three SEMs (**Fig. 1a-c**) based on three different aspects of stability (i.e., invariability, drought resistance, and invasion resistance) under the conditions of corresponding pairings of manipulated factors (**Fig. S1**), with the LAVAAN package (Rosseel 2012). Please see **Lines 527-537 and 546-554**, and **Fig. S1**.

5. This study focused on plant and faunal community composition and compositional stability as mediators in the effects of plant species richness on ecosystem multifunctionality. As such, I am curious about which species or species groups are important to stability of ecosystem multifunctionality. This point is not discussed well in the Discussion.

To address the Reviewer's question, we estimated the relative contribution of plant species (or faunal taxa at the order level) to the relationship between community stability and ecosystem multifunctional stability (invariability, drought resistance, or invasion resistance). The contribution of plant species (or the 2-, 3-, and 4-species random combinations) to the relationship between plant community stability and ecosystem multifunctional stability (invariability, drought resistance, or invasion resistance) was estimated by conducting following procedures:

1) The original relationship between the stability (invariability, drought resistance and invasion resistance) of original plant community (with no plant species removed) and ecosystem multifunctionality was calculated as the correlation (Pearson's R) between their pairwise similarities (**Fig. S1**).

2) We randomly removed 1 or a group of (random combinations of 2, 3 or 4 species from 16 plant species) plant species from the original community to imitate the null plant community. Then, the null relationship between the stability (invariability, drought resistance and invasion resistance) of null plant community and ecosystem multifunctionality was calculated.

3) Change (decrease) of the relationship between the community stability and ecosystem multifunctional stability after the random remove of plant species (or their random combinations) was calculated as the difference between the original relationship and the null relationship.

4) The relative contributions of plant species (or their random combinations) were calculated as the percentage decrease of community-multifunctional stability relationship after the remove of plant species, compared to the original relationship.

The same estimation procedures were also applied to litter/soil faunal communities, but that the species remove procedure was based on faunal taxa at

the order level. Please see **Lines 961-984**.

Based on these analyses, we found that the positive relationship between the stability (all aspects) of community (plant, soil fauna, and litter fauna) and multifunctionality decreased with the remove of plants species (or faunal orders) from the original community, and the level of such decrease increased with the number of plants species (or faunal orders) removed (**Fig. S3**), demonstrating that the co-working and interaction among multiple plants (or fauna) are important for the maintenance of multifunctional stability, despite that some dominant plant species (*Patrinia scabiosifolia*, *Artemisia stolonifera*, and their combinations with others) or faunal orders (soil fauna: *Posuromorpha* and *Trombidiformes*; litter fauna: *Trombidiformes* and *Araneae*) accounted for relatively high level of contributions (**Fig. S4**). We also found that the higher the mean abundances of the plant species or faunal taxa that drive the ecosystem functions, the greater their contribution to multifunctional stability (**Fig. S5a-c**). In addition, the contribution of plant species (or their combinations) to the community-multifunctional stability decreased with the mean coefficient of variance (CV) of biomass, while those of soil and litter faunal orders (or their combinations) decreased with the mean CV of faunal abundance (**Fig. S5d-f**), suggesting that the stability of dominant plant species' biomass against disturbances (spatial stochasticity, drought and invasion) while the asynchronous actions of faunal taxon that drive the dynamic of ecosystem functions were also key to the community-multifunctional stability relationship. Such results and discussion have been add to the discussion section. Please see **Lines 254-277** and **Fig. S3-5**.

Reviewer #2 (Remarks to the Author):

The authors collected extensive data for three years to investigate the effects of plant richness, drought and invasion, and their indirect effects on multifunctional stability. The data is valuable. However, I have major concerns on the experiments and statistical framework which might lead to false conclusion. I hope the authors can address them and improve the current work.

We thank the Reviewer for the helpful comments on the insufficient description for the experimental design and statistical analyses, which helped to increase the logic and soundness of the current study.

Major concerns:

1. In my opinions, the native species selection is always biased based on availability and easiness of experiment setup. However, the selection of native species can affect the results and interpretation greatly. Hence, the reason of the selection cannot just be

they are common species there. This would be the first flaw of the study. The authors need to document the attributes and functional traits of native species and their own interactions. The selection criteria should be included. Moreover, how these species could interact with invasive species? The invasive species might not be “invasive” enough for some native species. The document of the interaction between the invasive and native species is weak.

Thanks for the Reviewer’s questions. Sixteen locally frequent native plant species in the barren mountain areas (around Taizhou University, Zhejiang, China) invaded by the exotic plant *Symphyotrichum subulatum* (Hong and Blackmore 2013) were selected as the native species pool. These species were chosen because they spanned the dicotyledon plant taxonomy (including 7 Orders, 10 Families and 14 Genus, in the Class *Magnoliopsida*), differed widely in their functional traits (related to height, life form, dominance in local communities, and leaf habit) (**Table S3**), and were occasionally found to be associated with the invasive species *Symphyotrichum subulatum* (Hong and Blackmore 2013) in the local secondary-succession communities. With this species pool, we were able to imitate the locally-natural, spatially-stochastic, compositionally-ruderal, and functionally-varied plant community (Medvigy *et al.* 2019), which is a typical attribute of the secondary-succession communities, in the local barren mountains invaded by the exotic plant *Symphyotrichum subulatum*. The attributes and functional traits of these species were included in **Table S3**, and further information about the criteria for the selection of the 16 plant species was detailed in the Methods section. Please **Lines 383-394** and **Table S3**.

As to the interaction between native and invasion species, it is difficult to elaborate all the mechanisms. Instead, we decided to address the key points in relation to our study design briefly in the method section. Specifically, the invasive species can interact with the native species via, for example, competing for space and resources (Booth *et al.* 2003; Zhu *et al.* 2015), and changing soil faunal, bacterial or fungal microbiomes (Cavieres 2021; Fargione and Tilman 2005; Shen *et al.* 2021). Please see **Lines 412-415**.

2. The experiment is impressive. However, following the comment above, random mixtures of 2, 4, or 8 species can introduce different levels of noise into your work. The authors did not implement any correction for removing the noise. Moreover, there are 10, 10, or 9 distinct assemblages. The authors did not state the reason why assemblages were not designed as 10, 10, or 10.

Thanks for the Reviewer’s question. We agree with the Reviewer that random mixtures will introduce different level of noise as the number of full combinations of 2, 4, or 8 species from the 16 species pool are different (and it is impossible to manipulate all combinations). But this different level of random noise among

different plant species richness (1, 2, 4, and 8 species) treatments didn't influence our analyses, because the core analyses (SEMs) in relation to plant species richness in the current work were conducted by treating plant species richness gradient as a single influence factor, and thus didn't involve the comparison among different plant richness treatments which may be dependent on the noise within different plant richness treatments. Because of the limited space, we designed 9 distinct assemblages for the highest-diversity (8 plant species) treatment instead of 10, but this didn't influence our analyses on the effects of plant richness on the stability of community and multifunctionality for the similar reason as addressed above.

3. The data analyses are not appropriate. The authors introduced the Mantel test but lacked a clear statistical framework specifying the null and alternative hypotheses. For example, the tests can be flawed in the presence of autocorrelation and return erroneously low p-values (Reference: doi: 10.1111/2041-210x.12018)

Thanks for the Reviewer's question. The null hypotheses (H0) were that the independent factors plant richness, drought, and invasion, have no direct/indirect effects (effect = 0) on multifunctional stability. Based on H0, permutation procedure was conducted by permuting the index of dependent factors (both columns and rows of a symmetric matrix) to gain null models and null effects. *P* values (probability of H0 acceptance) were calculated as the percentage of observed positive (or negative) effect that was greater (or less) than the null effects (such information had been addressed in the Methods section. Please see **Lines 557-564**). Mantel (and also Mantel-like) permutation tests are widely used in the scientific community and while they are not well-grounded statistical methods (lacking strict statistical distribution for null hypotheses), no computer program implementing alternative methods have been developed (Guillot and Rousset 2013).

As to the Reviewer's doubt on the applicability of the Mantel-like permutation test for the current study, the key lies in the autocorrelation in both independent and dependent symmetric matrices (e.g., geographical distance, phenotypic distance, and genetic distance, which are common in evolutionary biology studies) that leads to erroneous *P* values as mentioned by Guillot and Rousset (2013). Despite the lack of strict statistical distribution for null hypotheses, Mantel permutation test is valid if either *x* or *y* is non-autocorrelated (Guillot and Rousset 2013). Based on this and the fact that, 1) there are no autocorrelation for the stationary independent variables (plant richness, drought and invasion) designed and assessed in this study, and 2) the manner these three independent variables were incorporated into SEMs and associated with the dependent variables (community or multifunctional stability) was not based on their pairwise differences/similarities as those applied for the tests between two (or multiple) distance matrices using Mantel (or partial Mantel) method but based on the real

identities/values of the independent factors in association with the pairwise similarity of the dependent variables (**Fig. S1**; please also refer to **Lines 527-537 and 546-554**), we are confident about the appropriateness of the permutation test applies for the analyses in current study.

In addition, to further eliminate the Reviewer's concern about the 'erroneously low p-values' we further build the same SEMs using the traditional method by calculating similarity in a strict one-to-one correspondence manner according to Baert *et al.* (2016). The results suggested that the multivariate-based SEMs (**Fig. 2b, c**) generated comparable levels of statistical significances of effects compared to the univariate-based ones (**Fig. S2b, c**), except for the effects of plant richness on plant community stability (see **Lines 229-235** for further explanations), suggesting that the Mantel-like permutation test did not result in erroneously low *p*-values for the current study. Please refer to **Lines 222-238** and **Fig. S2**.

Given the reasons above, I would not be so confident with the results. Indeed, the results are rather intuitive and match our common understanding of community ecology. I wonder what the novel or most important findings are here.

Biodiversity's (e.g., aboveground plant richness) and disturbances' (e.g., drought and exotic plant invasion) controls on ecosystem dynamics are central focus of ecological research aimed at maintaining and increasing the ecosystem (multi)functionality, in the face of global change. Despite the well-recognition of the intimate linkages between biological communities and ecosystem functions, it remains unknown whether and how biological community, or more specifically community compositional stability, mediates the stability of ecosystem multifunctionality in response to biodiversity loss and multiple disturbances. The present study digs into this novel and complex topic, with the aim to advance our mechanistic understanding of the stabilizing effect of plant diversity as well as destabilizing effects of drought/invasion on ecosystem multifunctional stability. Maybe we had not organized the discussion section properly, such that led the Reviewer to confuse about the main results and conclusion of this study. Thus, we re-addressed the novelty and the main results of the current study in the Conclusion section. Please see **Lines 364-380**.

The section of Discussion is disappointing because the authors only compared some of their findings to others. The scope is rather narrow. Authors also failed to document the caveats and potential improvement of the study.

Thanks for the Reviewer's comment. After careful check throughout the discussion section, we deleted some redundant sentences for clarity (Please see

Lines 222-238, 244, 293, 321-322, 330 and 351-352), and added some discussions on the contributions of specific plant species and faunal taxa groups to the community-multifunctional stability relationship (**Lines 254-277**) and the comparative insight into the strengths of the multivariate-based SEMs over that of the univariate-based ones (**Lines 222-238**) in the Discussion section, as well as the caveats and potential improvement of the study in the Conclusion section (Please see **Lines 364-380**).

- Baert, J.M., De Laender, F., Sabbe, K. & Janssen, C.R. (2016). Biodiversity increases functional and compositional resistance, but decreases resilience in phytoplankton communities. *Ecology*, 97, 3433-3440.
- Booth, M.S., Caldwell, M.M. & Stark, J.M. (2003). Overlapping resource use in three Great Basin species: implications for community invasibility and vegetation dynamics. *Journal of Ecology*, 91, 36-48.
- Cavieres, L.A. (2021). Facilitation and the invasibility of plant communities. *Journal of Ecology*, 109, 2019-2028.
- Fargione, J.E. & Tilman, D. (2005). Diversity decreases invasion via both sampling and complementarity effects. *Ecology Letters*, 8, 604-611.
- Guillot, G. & Rousset, F. (2013). Dismantling the Mantel tests. *Methods in Ecology and Evolution*, 4, 336-344.
- Hong, D. & Blackmore, S. (2013). *The Plants of China: A Companion to the Flora of China*. Cambridge University Press, United Kingdom.
- Medvigy, D., Wang, G., Zhu, Q., Riley, W.J., Trierweiler, A.M., Waring, Bonnie G. *et al.* (2019). Observed variation in soil properties can drive large variation in modelled forest functioning and composition during tropical forest secondary succession. *New Phytologist*, 223, 1820-1833.
- Naeem, S. & Li, S. (1997). Biodiversity enhances ecosystem reliability. *Nature*, 390, 507-509.
- Rosseel, Y. (2012). Lavaan: An R package for structural equation modeling. *Journal of Statistical Software*, 48, 1-36.
- Shen, C., Wang, J., He, J.-Z., Yu, F. & Ge, Y. (2021). Plant diversity enhanced soil fungal diversity and microbial resistance to plant invasion. *Applied and environmental microbiology*, 87, e00251-00221.
- Zhu, D.H., Wang, P., Zhang, W.Z., Yuan, Y., Li, B. & Wang, J. (2015). Sampling and complementarity effects of plant diversity on resource use increases the invasion resistance of communities. *PLoS One*, 10, e0141559.

Reviewer #2 (Remarks to the Author):

I think authors have improved the manuscript and addressed my concerns. I would recommend to accept the manuscript.